# Impact of CO_2_ on Pyrolysis Products of Bituminous Coal and Platanus Sawdust

**DOI:** 10.3390/polym11081370

**Published:** 2019-08-20

**Authors:** Ying Luo, Haoxi Ben, Zhihong Wu, Kai Nie, Guangting Han, Wei Jiang

**Affiliations:** 1Key Laboratory of Energy Thermal Conversion and Control of Ministry of Education, Southeast University, Nanjing 210096, China; 2State Key Laboratory of Bio-Fibers and Eco-Textiles, Qingdao University, Qingdao 266071, China

**Keywords:** CO_2_ atmosphere, tar research, NMR analysis, pyrolysis behaviors

## Abstract

Abundant studies have been completed about factors on the pyrolysis of coal and biomass. However, few articles laid emphasis on using CO_2_ as a carrier gas to explore the compositional changes of pyrolysis products in coal and biomass pyrolysis for industrial application and commercial value. The experiments on coal and biomass pyrolysis in N_2_ and CO_2_ using a horizontal tube furnace were conducted at 500 °C. The impact of introducing CO_2_ on the pyrolysis process of bituminous coal and Platanus sawdust was investigated. The nuclear magnetic resonance (NMR) spectra of tar and the characterizations of char including Brunner-Emmet-Teller (BET) measurements, scanning electron microscope (SEM), Fourier transform infrared (FT-IR) spectroscopy, and element analysis were studied. The findings in light of the experimental results show that introducing CO_2_ enhances the coal and biomass pyrolysis in a solid product by promoting the fracture of hydroxyl groups. It also promotes tar decomposition and the release of volatiles, which contribute to the occurrence of char with high porosity, pore volume, and specific surface. Furthermore, higher specific surface enhances the adsorption performance of char as active carbon. Simultaneously, CO_2_ promotes the increase of oxygen-containing aromatics especially the methoxy-containing aromatics, and the decrease of deoxygenated aromatic hydrocarbons in pyrolysis oils. In addition, the introduction of CO_2_ changes the amount of aliphatic compounds in various ways for the pyrolysis of coal and biomass. From a perspective of business, the changes in the composition of pyrolysis oil brought by CO_2_ may create new value for fuel utilization and industrial products.

## 1. Introduction

Among global energy resources, coal and biomass rank first and third, respectively, all over the world, while oil is the second largest energy resource [1]. Although the enormous energy generated by direct combustion of coal has positively accelerated the pace of human development, its leading role that includes global warming and air pollution has produced a negative influence on the environment [2,3]. As an independent thermochemical conversion process, pyrolysis technology is the initial stage step of combustion and reforming/gasification of coal [4] and of great importance for clean utilization of coal and biomass [5]. In addition, the research on pyrolysis products also has great development prospects.

The application of pyrolysis products is extensive and in line with the concept of sustainable development and green chemistry. The char is often used as an adsorbent [6] to adsorb pollutants and a catalyst [7] to catalyze experimental reactions. In addition, the solid product is also a soil conditioner to increase agricultural productivity. Pyrolysis oil is mainly utilized in the production of combustion heat, electricity production, fuel oil [8], and chemicals. Although the calorific value of pyrolysis oil is only about half of that of fossil fuel oil, the liquid product has the advantage of being easy to handle, transport, and store.

In recent years, pyrolysis of coal and biomass has become an interesting topic in the field of energy and chemical industry, both home and abroad [9,10,11,12]. The main factors affecting the pyrolysis of the two are catalyst, pyrolysis temperature, heating rate, atmosphere, gas flow rate, residence time, etc. [13]. Although a great number of investigations were accomplished on this pyrolysis behavior and probing deeply into the conditions in above fields, it is noteworthy that few articles laid emphasis on the atmosphere of pyrolysis and its conceivable effect. The surface structure, activity, and composition of char will be affected by the reaction gas, and the atmosphere surrounding char will also make a difference in the composition of tar. Therefore, it is of great significance to find out the pyrolysis mechanism of coal under various atmospheres. Moreover, some research results show that CO_2_ can promote the pyrolysis of biomass and the condensation of volatiles [14,15,16], while promoting tar cracking and inhibiting tar polymerization [17].

Duan et al. [18] investigated coal pyrolysis in CO_2_ and predicted two dominating impacts on the progress of coal pyrolysis. CO_2_ may change the yield and element of volatile compounds when they come from the pyrolysis of coal. Otherwise, CO_2_ is involved in the gasification reaction of char, then SO_2_/NOx precursors will form due to the participation of CO_2_. Farrow et al. [19] conducted the pyrolysis of sawdust and pinewood in a drop-tube furnace in both CO_2_ and N_2_ atmospheres at different temperatures. They found that the volatile yields can reach 99% in CO_2_ and over 96% in N_2_, which indicated that the biomass pyrolysis was actually a complete gasification process. Lee et al. [20] focused on the susceptibility to CO_2_ in pyrolysis of various biomasses and understood that CO_2_ enhanced the thermal cracking of volatile organic compounds (VOCs) evolved from the thermal degradation of biomass. Hanaoka et al. [21] investigated the gasification reaction of solid products under various atmospheres of N_2_/CO_2_/O_2_ at 800 °C. The result indicated that solid and gaseous substance after reaction increased when the pressure of production rose. Choi et al. [22] employed brown coal for pyrolysis in CO_2_ and comprehended that the quantity of benzene derivatives in the tar became smaller, inversely CO in gaseous products gradually increased. The results illustrated that the carbon element will shift from condensable compounds to incondensable CO due to the aid of CO_2_.

A large number of studies reported have been completed on the pyrolysis of coal and biomass [23,24]. Since the application of pyrolysis products has good prospects, the research on structural and compositional changes in char and pyrolysis oils under different atmospheres will help to enhance their commercial value and industrial application. As a result, the study of the impact of CO_2_ on bituminous coal and Platanus sawdust pyrolysis will be a fairly necessary foundation.

## 2. Materials and Methods

### 2.1. Sample Preparation

The sample of coal (NSC) was bituminous coal whose standard substance number was GBW11109 (n) produced in Shandong Metallurgical Research Institute (Jinan, China). The raw coal was naturally dried and crushed, then all passed through a 0.2 mm (80 mesh) sieve, and its particle size was less than 0.2 mm. The Platanus sawdust (Sipailou Campus, Southeast University, Nanjing, China) was selected as the sample of biomass (PS). The raw material was ground into 150–180 μm (80–100 mesh), then dried under vacuum drying oven at 333 K for 48 h to remove moisture. The proximate and ultimate analyses on a dry basis of the materials are exhibited in Table 1.

### 2.2. Tube Furnace Pyrolysis

The fast pyrolysis of NSC and PS was carried out at 500 °C under N_2_ and CO_2_ atmosphere in a tube furnace. The schematic diagram of the pyrolysis device is displayed in Figure 1. The initial side of the device was a gas supply unit that contained nitrogen and carbon dioxide tank, then it was connected to the assembled quartz tube through a gas pipe. The gas pipe was equipped with a pressure gauge for real-time pressure detection to avoid dangerous situations, such as overpressure. The mass flow controller ensured that the flow rate of gas was 500 mL/min and the air in the tube was discharged for 10 min. The quartz tube was 1 m long and has an inner diameter of 60 mm. Two quartz boats were placed inside to accommodate the sample. The quartz tube located horizontally in the tube groove inside the tube furnace, and the quartz tube can be electrically heated when the upper and lower cases of tube furnace were touched. The tube furnace can control the heating time and set the reaction temperature. The heating rate was set to 100 K/min, the char residence time was 20 min, and the gas residence was 3 min. It can also reflect the temperature and heating in the box in real-time during operation. The other end of the quartz tube was connected to the condensing tube. The two ports on the side of the condensing tube were connected to the freezing fluid circulation device. The inside of the device was used for cooling ethanol, and the condensation temperature was set to −10 °C. The outlet of the condensing pipe was used for receiving the condensed pyrolysis oil, and the generated gas was discharged. After that, the discharged gas passed through the drying pipe and entered the gas-sampling bag, then the gas exited into the atmosphere after the treatment. 

### 2.3. Physicochemical Analysis

FEI Inspect F50 scanning electron microscopy (SEM) (Thermo Fisher Scientific Inc., Waltham, MA, USA) was employed to examine the aperture structure of chars. ASAP 2020M (Micromeritics Inc., Norcross, GA, USA) was performed to obtain the results of specific surface area, pore diameter and pore volume for BET test. N_2_ adsorption isotherms and mercury porosimetry were conducted at 77.7 K. EURO EA3000 elemental analyzer (Eurovector Inc., Pavia, Italy) was used to explore the constituents of char and tar which utilized combustion to obtain hydrogen, carbon, sulfur, nitrogen contents, and confirm the oxygen content by mass difference.

### 2.4. Infrared Analysis of Char

Nicolet iS 10 FT-IR spectrometer (Thermo Scientific Inc., Waltham, MA, USA) was employed to investigate functional groups of the char. The range shown in the spectra was from 4000 to 700 cm^−1^, and the results were obtained from the average of multiple experimental data.

### 2.5. NMR Analysis of Tar 

AVANCE III HD 600MHz NMR spectrometer (Bruker Inc., Zurich, Switzerland) was used to acquire NMR spectral results of tar produced under different atmospheres. The samples, that weighed 100 mg, were dissolved in 500 μL deuterated dimethyl sulfoxide (DMSO-d_6_). The solvent presented seven peaks in ^13^C NMR spectra and the middle of peaks was at 39.51 ppm. The operation parameters were set to 1500 scans and 3 s pulse delay [25,26].

For ^1^H NMR, the tars were collected to dissolve in deuterated dimethyl sulfoxide (DMSO-d_6_) equally. The solvent presented five peaks and the middle of peaks was at 2.54 ppm. The operation parameters were set to 16 transients and 5 s pulse delay. All NMR results were processed by MestReNova v11.0 (Mestrelab Research, Santiago De Compostela, Spain).

## 3. Results and Discussion

### 3.1. Product Yield of Coal and Biomass Pyrolysis

The histograms of NSC and PS pyrolysis under N_2_ atmosphere and CO_2_ atmosphere are shown in Figure 2. The average values of solid and liquid yield were 73.33% and 15.73% in NSC, identically 23.16% and 43.36% in PS under N_2_ atmosphere. The error bars were less than 5%. It is clear that the pyrolysis yield varies under different atmospheres. The yield of char of NSC decreased by 3.21%, and similarly, PS declined by 2.17%, compared to N_2_ atmosphere. The yield of char produced in CO_2_ decreased slightly on account of the CO_2_ gasification of carbon:(CO_2_ + C → 2CO)(1)

The reaction is mild and the reduction of char is limited. There will be little distinct difference in weight loss of char under various atmospheres below 700 °C for the pyrolysis of coal. The coal pyrolysis yield was pretty low, and was lower than the proximate analysis yield, which also seems to mean that the temperature was too low for complete pyrolysis of NSC. The tar yield of NSC simultaneously decreased by 6.24% and PS declined by 11.58% in CO_2_, which indicates that CO_2_ reacts with tar to cause decomposition of tar:(tar + CO_2_ → CO + H_2_ + CH_4_+hydrocarbon (C_2_, C_4_, ...))(2)

According to the law of conservation of mass, it can be concluded that the gas production of NSC and PS increased, compared with N_2_ atmosphere. There are several speculations about the reaction between gases, such as vapor shift reaction:(CO + H_2_ → H_2_O + CO)(3)
and methane reforming reaction:(CO_2_ + CH_4_ → 2CO + 2H_2_)(4)

From the two graphs of the results of both biomass and coal pyrolysis, it can be seen that CO_2_ has a greater impact on the pyrolysis of tar in biomass compared with coal and produces more gas under these conditions. However, the specific component changes need further analysis and exploration.

### 3.2. Effect of CO_2_ on Char Properties

#### 3.1.1. Functional Groups of Char Structure

The Fourier transform infrared spectroscopies of char which reacted at 500 °C in N_2_ and CO_2_ are compared in Figure 3 to find out the impact of CO_2_ gasification reaction. In Figure 3, it can be seen that the two have very similar structural features. The absorbance at 3426 cm^−1^ (NSC) and 3414 cm^−1^ (PS) assigned to the hydroxyl group under CO_2_ atmosphere was slightly weaker than that in N_2_. Rare gap was observed in the peaks at 2911 cm^−1^ (NSC) and 2920 cm^−1^ (PS) representing –CH_2_– antisymmetric stretching vibration and at 2854 cm^−1^ (PS) representing –CH_2_– symmetric stretching vibration. The weaker absorbance manifests that the number of hydroxyl groups was decreasing, and CO_2_ promoted the cleavage of hydroxyl groups into volatiles during the pyrolysis of the raw materials, which suggests that CO_2_ gasification of carbon enhances the cleavage of the hydroxyl groups.

The –C=C– bands [27] in aromatic ring (1577 cm^−1^ (NSC) and 1556 cm^−1^ (PS)), aliphatic and cyclic ether C–O bands (1035 cm^−1^ (NSC) and 1170 cm^−1^ (PS)) and aromatic ring out-of-plane bending C–H (880–680 cm^−1^) [28,29] decreased gradually, which is a sign that plenty of hydroxyl groups and aliphatic compounds evolve into condensable matter.

#### 3.1.2. Characterization of Char Surface

##### SEM Analysis

As shown in Figure 4, the surface structure of the chars produced by NSC pyrolysis under atmospheres of N_2_ and CO_2_ differs. Figure 4a shows that there are a number of large or small nucleate particles, which vary in shape but are relatively concentrated, mainly due to inorganic components [30]. This is consistent with the structure of coal proposed by Haenel [31]. Different from the pyrolysis of coal under N_2_ atmosphere, the pyrolytic carbon from CO_2_ shown in Figure 4b is dispersed and loose. It seems that the CO_2_ gasification of the nascent char happens on the surface. Since the reaction temperature is not high, the activation energy required for the reaction cannot meet the requirements, the change is not obvious. The result is in line with the conclusion in Figure 2 which shows that two various chars produced in N_2_ and CO_2_ have similar mass. A more accurate analysis of carbon surface parameters requires the BET test of carbon, and the change of the two atmospheres can be observed from the difference of pore size and surface area.

Apparently, the morphological structure of solid products obtained from pyrolysis of biomass samples under N_2_ and CO_2_ atmosphere differs and the SEM pictures of char generated from pyrolysis of PS are exhibited in Figure 4. In Figure 4c, the fibrous structure of biochar under N_2_ atmosphere is noticeable, while porous cavities are also visible from the images of morphology. Additionally, the fibers begin to break, and the fragments are numerous and disorderly, which is perhaps the result of fast release of condensable substance during rapid pyrolysis [32]. For the biochar generated in CO_2_, the pictures in Figure 4d indicate an important difference in its morphological structure, which exhibits more broken and sharper vascular bundles and a great deal of fines, which can be ascribed to the reaction of carbon and carbon dioxide. Biochar generated from pyrolysis of PS in CO_2_ evolves into abundant small structures, which indicates that the morphology makes a great difference when the surrounding gas is CO_2_. The observation proves the impact of CO_2_ on char surface more directly and obviously comparing with the pyrolysis of NSC. 

##### BET Analysis

The parameters of pore structure of the chars under the atmospheres of N_2_ and CO_2_ are presented in Table 2. The release of gas from the reaction of char gasification during pyrolysis can open the old holes and produce new holes, which will lead to the expansion of closed pores of char [33] and contribute to the change in pore size distribution and specific surface area. 

When pyrolyzing the coal and biomass, the solid production will reduce gradually and the volatile production will grow oppositely. Higher BET surface area and total pore volume of the char appears under CO_2_ atmosphere other than in N_2_. The reaction of char gasification brings about a rise in pore volume, open porosity, and specific surface area of the solid product. 

In addition, CO_2_ promotes the decomposition of tar into small molecules, which inhibits the polymerization of tar and hinders the formation of secondary reaction carbon. Meanwhile, the secondary carbon that adheres to microspores leads to pore blockage and reduces BET specific surface area of carbon in the case of nitrogen.

##### Elemental Composition of Char

As can be seen in Table 3, coal and biomass have more carbon content in the pyrolysis of CO_2_, which increased by 1.99% and 1.81%, respectively. In contrast, the hydrogen content decreased by 0.02% and 0.05%, which indicates that there is little change or just an error. The amount of oxygen reduced by about 2%. It illustrates that CO_2_ has a significant role in enhancing the cracking of hydroxyl groups under the pyrolysis of coal and biomass. It also explains that the amount of carbon in char increased after the occurrence of CO_2_ gasification reaction of the nascent char. In addition, within the margin of error, there is almost no change in the content of nitrogen and sulfur both in the case of pyrolysis of NSC and PS. Compared with the ultimate analysis of biomass, the O and H content reduced a lot, which indicates that a large number of hydroxyl groups in PS are converted into volatiles after pyrolysis. Then the C and N content grew relatively and were higher than that of char under the pyrolysis of coal. The elemental composition of char for the reaction of coal kept constant with the ultimate analysis of NSC, which also manifests incomplete pyrolysis for bituminous coal at 500 °C.

As a result, the introduction of CO_2_ enhances the cracking of hydroxyl groups for biomass and coal. The char gasification reaction promotes the pyrolysis process of PS and NSC, which means lower char yield and a greater amount of volatiles. CO_2_ also makes a higher BET surface area, then helps expand total pore volume.

### 3.3. Impact of CO_2_ on Tar Components

#### 3.3.1. Analysis of ^1^H NMR Spectra

The integral regions of various functional groups in the tar produced by the pyrolysis of NSC and PS under different atmospheres are summarized in Appendix A. Table 4 lists the specific numericals of chemical shift ranges and integral peak area in ^1^H NMR spectra under N_2_ and CO_2_ atmosphere. The chemical shift range of 9.6–10.0 ppm was assigned to aldehydes and acids. Whether it was NSC or PS, the amount of pyrolytic oil in this region was extremely low in both atmospheres. Polycyclic aromatic hydrocarbon protons were assigned to 7.5–9.0 ppm, which shows that its composition has been greatly reduced by 9.79% in the pyrolysis oils of coal under CO_2_ atmosphere. At the same time, rthe single ring aromatic protons that were assigned to 6.0–7.5 ppm also decreased by 5.14% in the tar of NSC in CO_2_. The chemical shift region of ~4.0–5.0 ppm was assigned to protons on the aromatic hydroxyl group, and that of ~3.8 ppm was assigned to aromatics methoxyl protons. These components all rose in the atmosphere of CO_2_, especially the change of CH_3_–O–aromatic composition exceeded 10% in the coal pyrolysis. The amount of aliphatic methoxyl protons (~3.3 ppm) dropped from 10.41% (in N_2_) to 1.95% (in CO_2_) in the pyrolysis of NSC, and fell from 11.85% (in N_2_) to 8.77% (in CO_2_) in the tar of PS. CH_3_-aromatic compounds (~2.2 ppm) made up for a small part in hydrogen percentages, which was less than 6%. The protons on aliphatic methyl and methylene that were assigned to 0.0–2.0 ppm took up a large part in the pyrolysis oil. They varied inversely in the tar of NSC and PS under the two atmospheres.

Appendix A shows that the composition of the tar varies greatly. The downfield spectrum regions arising from the aldehydes and acids had less than 1% protons in both atmospheres. The methoxy-containing aliphatic group decreased in CO_2_, while the amount of the methyl/methylene-containing aliphatic group increased. It seems to show that CO_2_ can promote the demethoxylation of aliphatic hydrocarbons, which is in line with the observation of Tarves et al. [34] who found that fewer oxygen-containing structures and more deoxygenated compounds in the tar would generate surrounded by reactive gas atmospheres. Coal will form more polycyclic aromatic hydrocarbons during pyrolysis under N_2_ atmosphere, but it will be significantly reduced under CO_2_ atmosphere. It illustrates that CO_2_ has an effect of the oxidation and decomposition on PAH to some extent. Appendix A also shows that the amount of O-containing aromatic has increased (e.g., aromatic–OH, CH_3_–O–aromatic), while the number of single-ring aromatic hydrocarbons and methyl-containing aromatics has declined. It may be due to the fact that the reaction involving CO_2_ produces more water, which proves that CO_2_ has a certain oxidation effect on the aromatic ring chains.

Similar to the pyrolysis tar component of NSC, the acid and aldehyde functional group content in the PS pyrolysis process was extremely small, and the aliphatic compound amount was otherwise, more than 35%. In contrast to the pyrolysis oil of coal, the amount of polycyclic aromatic hydrocarbons in PS tar was not rich. At the same time, it increased a little under CO_2_ atmosphere. Another difference from the NSC pyrolysis was that the methyl/methylene-containing aliphatic content decreases under CO_2_ atmosphere. Appendix A also illustrates that the amount of O-containing aromatic has increased (e.g., aromatic–OH, CH_3_–O–aromatic), while the number of single-ring aromatic hydrocarbons and methyl-containing aromatics has declined. The result also explains the CO_2_ oxidation of aromatic hydrocarbons in the sawdust pyrolysis product under certain circumstances. In short, it can be observed that more protons of methoxy-containing aromatics and less aliphatic protons appear in the oil by the pyrolysis of biomass under CO_2_ atmosphere [35].

#### 3.3.2. Analysis of ^13^C NMR Spectra

The ^13^C NMR method was utilized to acquire more information of functional groups in the tars for further research. For ease of analysis, the chemical shift ranges and integration in the regions of components in pyrolysis oils were used to process the results [25] and the data are listed in Table 5. Representative ^13^C NMR spectra data for the tar obtained from the pyrolysis of NSC under CO_2_ and N_2_ atmosphere are described in Appendix A, then, the integration data for the pyrolysis tars of NSC and PS are summarized in Appendix A.

The figures show that the carbonyl group content (166.5–215.0 ppm) increased under CO_2_ atmosphere compared with that under N_2_ atmosphere under the pyrolysis of coal and biomass. It could be that CO_2_ was involved in the reaction of tar. The chemical shift range of 95.8−166.5 ppm was assigned to aromatic compounds which contained C–O bond (142.0–166.5 ppm), C–C bond (125.0–142.0 ppm), C–H bond (95.8–125.0 ppm). With the act of CO_2_, the total content of aromatic carbons rose in the NSC pyrolysis while decreased in the PS pyrolysis. In addition, aromatic C–H bond reduced by 5.27% in NSC and 7.46% in PS which is consistent with the results of ^1^H NMR analysis. The composition of levoglucosan accounted for a small part, less than 5% and even in the tar of PS less than 1%. Obviously, around 28% of carbons resonated in the region of 60.8–95.8 ppm manifesting that the tar of NSC pyrolysis contains abundant aliphatic C–O bond carbons. On the contrary, the same ingredient had just a proportion of 4% approximately. The methoxyl-aromatic bond was assigned to 55.2–60.8 ppm, and the variation tendency was also the same as the conclusion of ^1^H NMR spectrum. The amount of general aliphatic C–C bond (0.0–55.2 ppm) dropped from 31.98% (in N_2_) to 22.11% (in CO_2_) in the pyrolysis of NSC, but raised from 31.74% (in N_2_) to 40.56% (in CO_2_) in the tar of PS.

On the whole, for the pyrolysis of coal or biomass, the impact of CO_2_ on tar is obvious. It promotes the decomposition of tar and causes a great change in the composition, mainly in the variation of carbonyl, aromatic, and aliphatic content. The content of oxygen-containing aromatics, especially the methoxy-containing aromatics has increased, but the number of deoxygenated aromatic hydrocarbons becomes smaller and smaller. Meanwhile, the methoxyl-aliphatic compounds in the amount also go up both in the tar of NSC and PS pyrolysis.

#### 3.3.3. Elemental Composition of Tar

CHNO data of tars in NSC and PS pyrolysis under N_2_ atmosphere and CO_2_ atmosphere are listed in Table 6. As the most abundant element in whole tar, carbon content made a typical difference in the pyrolysis oil under atmospheres of N_2_ and CO_2_. It fell from 71.14% to 67.02% in the pyrolysis of PS while increased from 55.84% to 58.43% in that of NSC. The nitrogen content varied little when the experiment was conducted in CO_2_. In terms of mass equilibrium, the composition of oxygen and hydrogen reduced corresponding to a growth in the composition of carbon in the tar from the pyrolysis of NSC and raised in accordance with a fall in the content of carbon in PS pyrolysis oil, which is mainly the effect of the change in the amount of aromatic and aliphatic compounds.

## 4. Conclusions

Introducing CO_2_ enhances the pyrolysis of coal and biomass by facilitating the occurrence of char gasification and the cracking of hydroxyl groups. Pore diameter, pore volume, and BET surface area improve under the action of CO_2_ gasification reaction. Furthermore, higher specific surface enhances the adsorption performance of char as active carbon, which increases the industrial value of pyrolysis products. In addition, CO_2_ promotes the increase of oxygen-containing aromatics, especially the methoxy-containing aromatics and the decrease of deoxygenated aromatic hydrocarbons in pyrolysis oils. The amount of aliphatic compounds in the tar under CO_2_ atmosphere also gets changed simultaneously. CO_2_ has a certain oxidation impact on the components of tar. From the perspective of business, the changes in the composition of pyrolysis oil brought by CO_2_ may create new value for fuel utilization and industrial products.

## Figures and Tables

**Figure 1 polymers-11-01370-f001:**
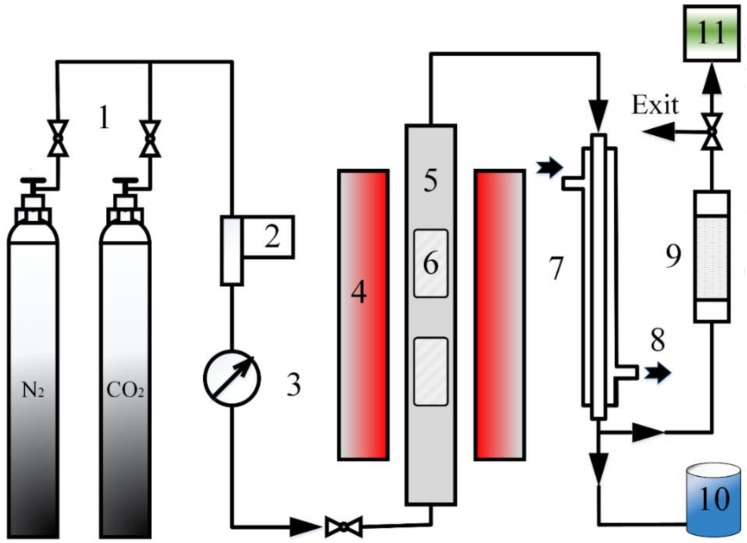
Schematic diagram of pyrolysis device formatting. 1: Gas supply unit, 2: Mass flow controller, 3: Pessure gauge, 4: Tube furnace, 5: Quartz tube, 6: Quartz boat, 7: Condensation tube, 8: Freezing fluid, 9: Quicklime dryer, 10: Liquid collector, 11: Gas-sampling bag.

**Figure 2 polymers-11-01370-f002:**
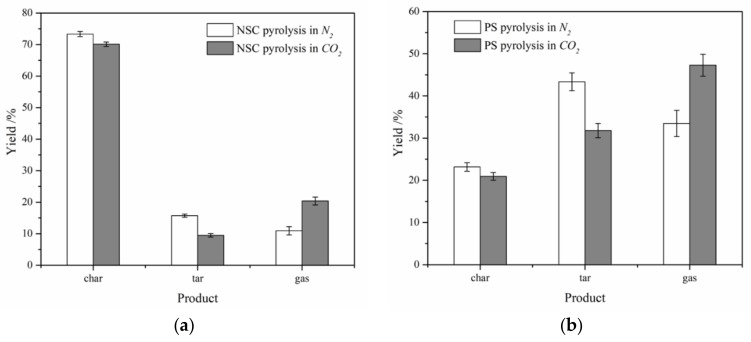
Yield distribution under N_2_ atmosphere and CO_2_ atmosphere: (**a**) NSC, (**b**) PS.

**Figure 3 polymers-11-01370-f003:**
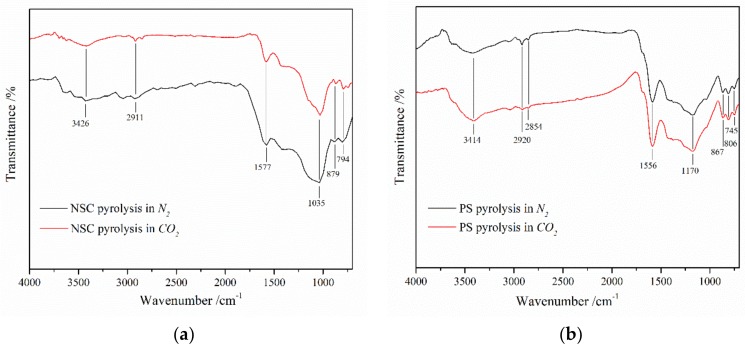
FT-IR of the char under N_2_ atmosphere and CO_2_ atmosphere: (**a**) NSC, (**b**) PS.

**Figure 4 polymers-11-01370-f004:**
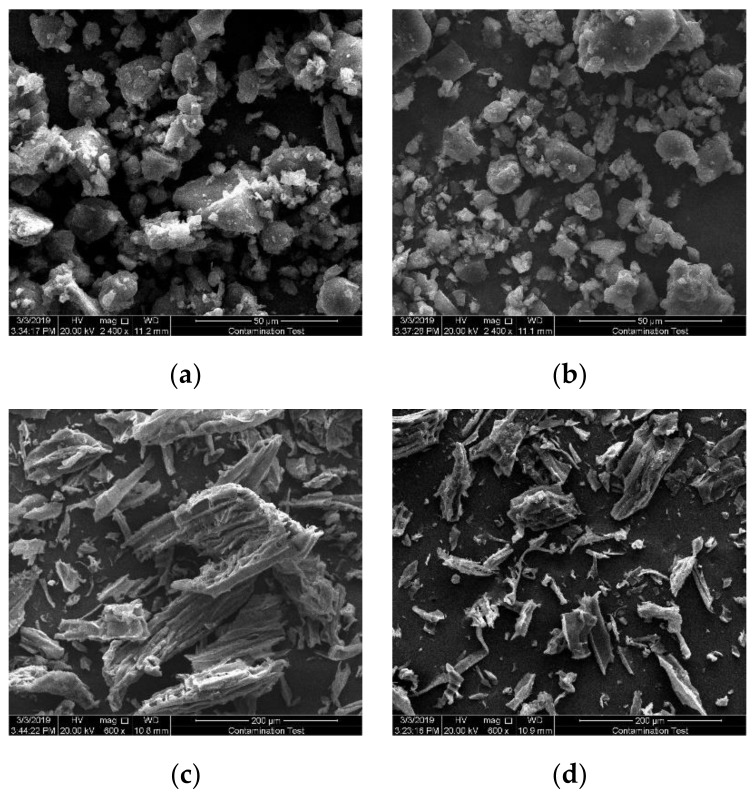
SEM image of char generated from the pyrolysis: (**a**) NSC in N_2_, (**b**) NSC in CO_2_, (**c**).PS in N_2_, (**d**) PS in CO_2_.

**Table 1 polymers-11-01370-t001:** Proximate and ultimate analysis of coal and biomass samples.

Sample	Proximate Analysis *w_d_*/%			Ultimate Analysis *w_d_*/%				
	Ash	Volatile	Fixed Carbon	C	H	O *	N	S
NSC	17.06	34.99	47.95	65.69	4.35	25.84	1.17	2.95
PS	1.46	80.57	17.97	51.08	6.43	42.42	0.07	--

* by difference.

**Table 2 polymers-11-01370-t002:** Pore structure parameters of the pyrolysis char under CO_2_ atmosphere and N_2_ atmosphere.

Sample Type	BET Surface Area A/(m^2^·g^−1^)	Average Pore Diameter d/nm	Total Pore Volume v/(cm^3^·g^−1^)
NSC(N2)	123.56	2.24	0.069
NSC(CO2)	152.41	2.15	0.082
PS(N2)	288.18	2.11	0.152
PS(CO2)	334.45	1.97	0.165

**Table 3 polymers-11-01370-t003:** Content of C, H, and N of char obtained from NSC and PS pyrolysis under CO_2_ atmosphere and N_2_ atmosphere they are cited.

Element	NSC(N_2_)	NSC(CO_2_)	PS(N_2_)	PS(CO_2_)
C *w*/%	64.86	66.85	80.23	82.04
H *w*/%	3.24	3.22	3.86	3.81
O *w*/%	28.44	26.27	15.72	13.98
N *w*/%	1.11	1.19	0.19	0.17
S *w*/%	2.35	2.47	--	--

**Table 4 polymers-11-01370-t004:** Hydrogen percentage based on the ^1^H NMR analysis of tar from NSC and PS under N_2_ atmosphere and CO_2_ atmosphere.

Assignments	Chemical Shift Ranges (ppm)	Hydrogen Percentages			
		NSC(N_2_)	NSC(CO_2_)	PS(N_2_)	PS(CO_2_)
–CHO, –COOH	9.6–10.0	0.27	0.14	1.21	0.04
(H–PAH)	7.5–9.0	11.66	1.87	1.69	2.96
(H-single ring aromatic)	6.0–7.5	8.49	3.35	0.11	0.31
(aromatic–OH, water)	~4.0–5.0	5.86	14.36	4.28	12.62
(CH_3_–O–aromatic, water)	~3.8	15.52	30.05	33.52	36.75
(CH_3_–O–aliphatic)	~3.3	10.41	1.95	11.80	8.77
(CH_3_–aromatic)	~2.2	5.24	1.17	3.85	2.83
(CH_2/3_–aliphatic)	0.0–2.0	42.55	47.11	43.54	35.72

**Table 5 polymers-11-01370-t005:** Carbon percentage based on the ^13^C NMR analysis of tar from NSC and PS under N_2_ atmosphere and CO_2_ atmosphere.

Functional Group		Integration Region (ppm)	Carbon Percentages			
			NSC(N_2_)	NSC(CO_2_)	PS(N_2_)	PS(CO_2_)
Carbonyl or Carboxyl bond		215.0–166.5	3.48	8.73	2.36	6.31
Aromatic C–O bond		166.5–142.0	0.92	1.29	1.72	4.95
Aromatic C–C bond		142.0–125.0	5.97	14.22	27.92	17.94
Aromatic C–H bond		125.0–95.8	17.63	12.36	17.25	9.79
Levoglucosan		C1 102.3, C2 72.0	3.99	3.17	0.16	0.20
		C3 73.7, C4 71.7				
		C5 76.5, C6 64.9				
Aliphatic C–O bond		95.8–60.8	28.48	28.78	3.29	4.00
Methoxyl-Aromatic bond		60.8–55.2	11.56	12.51	14.70	16.46
Aliphatic C–C bond	General	55.2–0.0	31.98	22.11	32.74	40.56
	Methyl-Aromatic	21.6–19.1	3.09	3.04	4.25	2.29
	Methyl-Aromatic at ortho position of a hydroxyl or methoxyl group	16.1–15.4	0.87	1.05	1.17	0.96

**Table 6 polymers-11-01370-t006:** Content of C, H, O, and N of tar obtained from NSC and PS pyrolysis under CO_2_ atmosphere and N_2_ atmosphere.

Element	NSC (N_2_)	NSC (CO_2_)	PS (N_2_)	PS (CO_2_)
C *w*/%	55.84	58.43	71.14	67.02
H *w*/%	8.47	7.23	9.32	8.97
O *w*/%	35.40	33.96	17.98	22.64
N *w*/%	0.29	0.38	1.56	1.37

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
