# Peer review of "Impact of CO2 on Pyrolysis Products of Bituminous Coal and Platanus Sawdust"

_polymers, 2019, doi:10.3390/polym11081370_

Round 1

Reviewer 1 Report

The present paper is an investigation about the adoption of CO2 as carrier gas during the coal and biomass pyrolysis. The authors well described all the characteristics of chars and the comparison with those obtained by working under nitrogen gas. 

I have some suggestions to ameliorate the reading: the reactions could be placed in separate lines; in Table 1 all parameters have to be specified.

Author Response

Point 1: The reactions could be placed in separate lines.

Response 1: We are thankful for the reviewer's comment. The format of reactions has been modified.

Modifications:

The yield of char produced in CO2 decreased slightly on account of the CO2 gasification of carbon:

                                       CO2+C→2CO                                                  (1)

The reaction is mild and the reduction of char is limited, which prove the statement of Gao et al. [11]. There will be little distinct difference in weight loss of char under various atmospheres below 700℃ for the pyrolysis of coal. The coal pyrolysis yield was pretty low, and was lower than the proximate analysis yield, which also seems to mean that the temperature was too low for complete pyrolysis of NSC. The tar yield of NSC simultaneously decreased by 6.24% and PS declined by 11.58% in CO2, which indicates that CO2 reacts with tar to cause decomposition of tar:

                   tar+CO2→CO+H2+CH4+hydrocarbon (C2, C4, ...)              (2)

According to the law of conservation of mass, it can be concluded that the gas production of NSC and PS increased compared with N2 atmosphere. There are several speculations about the reaction between gases, such as vapor shift reaction:

                                       CO+H2→H2O+CO                                           (3)

and methane reforming reaction:

                                      CO2+CH4→2CO+2H2                                       (4)

Point 2: In Table 1 all parameters have to be specified. 

Response 2: We thank the reviewer’s advice about statement. All parameters in Table 1 have been specified.

Added part:

A: ash; V: volatile; FC: fixed-carbon; C: carbon; H: hydrogen; O: oxygen; N: nitrogen; S: sulfur.

*by difference.

Reviewer 2 Report

The paper by Luo et. al describes the effect of CO2 on pyrolysis products of coal and sawdust, specifically "tar". They also compare their characterizations with pyrolysis products from a N2 atmosphere. It is a topic of practical importance. The paper is well-written, it is rich in data and referencing is quite good. Most statements are supported by the data and authors do not generally over interpret or oversell their work.

However, I have certain minor observations/corrections (attached) and some major concerns on the overall purpose of the work and the way it is presented in its actual form. The effect of pyrolysis media on pyrolysis products (char and oil) have been studied extensively, individually with biomass or coal and both. Infact, authors cite some of these references: Zhang et al. 2011, Gao et al. 2013. In my opinion, at best, this work is incremental and does not describe or point to any new findings. If authors are willing to revise it to highlight novelty, it can be accepted for publication. Otherwise, it is not.

Comments:

1.The goal of this the work is not clear, if effect of CO2 on "tar characteristics" was the goal, I don't think that is reflected in the title  and discussion of the manuscript.

2. The novelty could be highlighted by briefly describing the state of the art with regards to pyrolysis products.

3. Depending on which type of atmosphere used (CO2 vs N2), you can have pyrolysis products with different chemistry's. What is the economic significance of the findings of this study? Maybe you may want to include a
couple of sentences in your "abstract & Conclusion” on this.

Author Response

Point 1: The goal of this the work is not clear, if effect of CO2 on "tar characteristics" was the goal, I don't think that is reflected in the title  and discussion of the manuscript.

Response 1: Thanks for the reviewer's comment. The purpose of this article is to explore the effects of CO2, mainly on solid and liquid products. We have modified part of content in abstract & introduction to emphasize this goal.

Modified part in Row14-16:

However, few articles laid emphasis on using CO2 as a carrier gas to explore the compositional changes of pyrolysis products in coal and biomass pyrolysis for industrial application and commercial value.

Modified part in Row74-78:

Since the application of pyrolysis products has good prospects, the research on structural and compositional changes in char and pyrolysis oils under different atmospheres will help to enhance their commercial value and industrial application. As a result, the study of impact of CO2 on bituminous coal and platanus sawdust pyrolysis will be a fairly necessary foundation.

Point 2: The novelty could be highlighted by briefly describing the state of the art with regards to pyrolysis products.

Response 2: We are grateful for the advice of the reviewer. The state of art with regards to pyrolysis products is significant to show novelty. We added a paragraph in the introduction to implement it.

Added paragraph in Row40-47:

The application of pyrolysis products is extensive and in line with the concept of sustainable development and green chemistry. The char is often used as an adsorbent [6] to adsorb pollutants and catalyst [7] to catalyze experimental reactions. In addition, the solid product is also a soil conditioner to increase agricultural productivity. Pyrolysis oil is mainly utilized in the production of combustion heat, electricity production, fuel oil [8] and chemicals. Although the calorific value of pyrolysis oil is only about half of that of fossil fuel oil, the liquid product has the advantage of being easy to handle, transport and store.

[6]Fu, Y.; Shen, Y.; Zhang, Z.; Ge, X. & Chen, M. Activated bio-chars derived from rice husk via one- and two-step KOH-catalyzed pyrolysis for phenol adsorption. Science of The Total Environment 2019, 646, 1567–1577.

[7]Benedetti, V.; Ail, S. S.; Patuzzi, F. & Baratieri, M. Valorization of Char From Biomass Gasification as Catalyst Support in Dry Reforming of Methane. Frontiers in Chemistry 2019, 7.

[8]Bharathy, S.; Gnanasikamani, B. & Radhakrishnan Lawrence, K. Investigation on the use of plastic pyrolysis oil as alternate fuel in a direct injection diesel engine with titanium oxide nanoadditive. Environmental Science and Pollution Research 2019.

Point 3: Depending on which type of atmosphere used (CO2 vs N2), you can have pyrolysis products with different chemistry's. What is the economic significance of the findings of this study? Maybe you may want to include a
couple of sentences in your "abstract & Conclusion” on this.

Response 3: We thank the reviewer for the suggestion very much. The economic significance of the findings is necessary. We add some contents in our "abstract & conclusion".

Added part in Row23-24, 27-29, 402-404, 407-409:

Furthermore, higher specific surface enhances the adsorption performance of char as active carbon, which increases the industrial value of pyrolysis products.

From a perspective of business, the changes in the composition of pyrolysis oil brought by CO2 may create new value for fuel utilization and industrial products.

Point 4: In fact, authors cite some of these references: Zhang et al. 2011, Gao et al. 2013. In my opinion, at best, this work is incremental and does not describe or point to any new findings.

Response 4: We are thankful for the comments. The two articles have been replaced by other articles.

Replaced articles in Row62-67:

Farrow, T. S. et al. [19] conducted the pyrolysis of sawdust and pinewood in a drop-tube furnace in both CO2 and N2 atmospheres at different temperatures. They found that the volatile yields can reach 99% in CO2 and over 96% in N2, which indicated that the biomass pyrolysis was actually a complete gasification process. Lee, J. et al [20] focused on the susceptibility to CO2 in pyrolysis of various biomass and understood that CO2 enhanced the thermal cracking of VOCs evolved from the thermal degradation of biomass.

[19]Farrow, T. S.; Sun, C. & Snape, C. E. Impact of CO2 on biomass pyrolysis, nitrogen partitioning, and char combustion in a drop tube furnace. Journal of Analytical and Applied Pyrolysis 2015, 113, 323–331.

[20]Lee, J.; Oh, J.-I.; Ok, Y. S. & Kwon, E. E. Study on susceptibility of CO2 -assisted pyrolysis of various biomass to CO2. Energy 2017, 137, 510–517.

Reviewer 3 Report

The paper is interesting and could be published but requires a careful review by the authors especially regarding the English as some sentence are difficult to understand:

Ex:

row 44: What do the authors by "...ingredient of tarr"?

rows 46-47:"does volatile precipitation" mean condensation of volatiles?

rows 58-59: I think that "....when the production pressure boasted" sould be replace with something else;

Row 77, Table 2, Proximate analysis: A, V and FC should be explained;

Row 169, Pay attention to the name of the author in the refference [28]

Author Response

Point 1: Row44: What does the authors by "...ingredient of tar"?

Response 1: We thank reviewer’s advice about statement. The sentence has been modified.

Modification:

The surface structure, activity and composition of char will be affected by the reaction gas, and the atmosphere surrounding char will also make a difference on the composition of tar.

Point 2: Rows46-47:"does volatile precipitation" mean condensation of volatiles?

Response 2: We are thankful for the reviewer's comment. The sentence has been modified.

Modification:

Moreover, some research results show that CO2 can promote the pyrolysis of biomass and the condensation of volatiles.

Point 3: Rows58-59: I think that "...when the production pressure boasted" should be replaced with something else.

Response 3: We appreciate the reviewer's Suggestion. The sentence has been modified.

Modification:

The result indicated that solid and gaseous substance after reaction increased when the pressure of production rose. 

Point 4: Row77, Table 1, Proximate analysis: A V and FC should be explained. 

Response 4: We are grateful for the advice of reviewer. The proximate analysis: A V and FC in Table 1 have been explained.

Added part:

A: ash; V: volatile; FC: fixed-carbon; C: carbon; H: hydrogen; O: oxygen; N: nitrogen; S: sulfur.

*by difference.

Point 5: Row169, Pay attention to the name of the author in the reference [28]

Response 5: Thanks for the reviewer's correction. I am very sorry for misspelling the author's name, and I have made a modification.

Modification:

This is consistent with the structure of coal proposed by Matthias w. haenel [28].

[28] Haenel, M. W. Recent progress in coal structure research. Fuel 1992, 71, 1211–1223.